# Netrin Family Genes as Prognostic Markers and Therapeutic Targets for Clear Cell Renal Cell Carcinoma: Netrin-4 Acts through the Wnt/β-Catenin Signaling Pathway

**DOI:** 10.3390/cancers15102816

**Published:** 2023-05-18

**Authors:** Shuai Ke, Jiayu Guo, Qinghua Wang, Haoren Shao, Mu He, Tao Li, Tao Qiu, Jia Guo

**Affiliations:** 1Department of Urology, Renmin Hospital of Wuhan University, Wuhan 430060, China; keshuai@whu.edu.cn (S.K.); drguojy@whu.edu.cn (J.G.); wqh1996@whu.edu.cn (Q.W.); 2017302180212@whu.edu.cn (H.S.); 2018305231077@whu.edu.cn (M.H.); 2019305231027@whu.edu.cn (T.L.); 2The Department of Organ Transplantation, Renmin Hospital of Wuhan University, Wuhan 430060, China

**Keywords:** clear cell renal cell carcinoma, netrin, prognosis, therapeutic approaches, Wnt/β-catenin

## Abstract

**Simple Summary:**

The current outcomes for early diagnosis and late treatment of clear cell renal cell carcinoma (ccRCC) are unsatisfactory. This study aimed to determine the diagnostic and prognostic value of Netrin family genes for ccRCC and to explore new therapeutic targets. We systematically analyzed the expression profile of Netrin family genes in ccRCC using a bioinformatics approach and identified three genes, including NTN4, as potential biomarkers for ccRCC prognosis. In vitro and in vivo experiments confirmed that NTN4 regulates the growth and invasion of ccRCC cells by inhibiting the Wnt/β-catenin signaling pathway. Targeting this pathway through NTN4 may be a promising strategy for ccRCC treatment. The results of this study may have important implications for developing personalized cancer treatment strategies and identifying new biomarkers for ccRCC.

**Abstract:**

Clear cell renal cell carcinoma (ccRCC, or KIRC) is the most common type of kidney cancer, originating within the renal cortex. The current outcomes for early diagnosis and late treatment of ccRCC are unsatisfactory. Therefore, it is important to explore tumor biomarkers and therapeutic opportunities for ccRCC. In this study, we used bioinformatics methods to systematically evaluate the expression and prognostic value of Netrin family genes in ccRCC. Through our analysis, three potential biomarkers for ccRCC were identified, namely NTNG1, NTNG2, and NTN4. Moreover, we performed in vitro and in vivo experiments to explore the possible biological roles of NTN4 and found that NTN4 could regulate ccRCC development through Wnt/β-catenin signaling. We elucidate the molecular mechanism by which NTN4 modulates β-catenin expression and nuclear translocation to inhibit ccRCC progression, providing a new theoretical basis for developing therapeutic targets for ccRCC. Thus, we suggest that Netrin-related studies may offer new directions for the diagnosis, treatment, and prognosis of ccRCC patients.

## 1. Introduction

Renal cell carcinoma (RCC) has been increasing in incidence and is one of the most common malignancies of the genitourinary system, accounting for 90% of all renal malignancies [1]. Partial or radical nephrectomy is a curative option for patients with early or localized kidney cancer, but early diagnosis is difficult [2]. About 30% of kidney cancer patients are diagnosed with metastasis, and this group will still have metastasis after surgical treatment [3].

The three main tissue types of kidney cancer are papillary renal cell carcinoma, chromophobe renal cell carcinoma, and ccRCC, with ccRCC accounting for more than 70% of cases [1]. Therefore, elucidating the molecular mechanisms of ccRCC and discovering new biomarkers and therapeutic targets are essential for early diagnosis, treatment, and improved prognosis of ccRCC patients.

Netrins are a group of proteins conserved across animals and humans. The protein family comprises six secreted proteins, Netrin-1 (NTN1), Netrin-3 (NTN3), Netrin-4 (NTN4), Netrin-5 (NTN5), Netrin-G1 (NTNG1), and Netrin-G2 (NTNG2). They exhibit structural homology to the extracellular matrix protein laminin, with the amino-terminal two-thirds of NTN1 and NTN3 having sequences homologous to those in the laminin-γ1 chain [4,5]. In contrast, the amino-terminal structural domains of NTNG1, NTNG2, and NTN4 are homologous to those in the laminin-β1 chain [6,7,8]. Functionally, Netrins influence neurological and tumor progression. For example, NTN1 has a neuroprotective role in Parkinson’s patients [9], whereas, in colorectal tumors, apoptosis-dependent receptor deletion due to Netrin-1 receptor deficiency exacerbates tumor progression [10]. Upregulation of NTN1 also promotes the growth of various tumor cells, such as melanoma, breast cancer, and liver cancer [11,12,13]; the function of NTNG1 is to increase the migration of ovarian cancer cells [14]. In glioblastoma cells, NTN4 activates the mTOR signaling pathway and promotes cell proliferation. However, in rectal tumors, NTN4 exhibits anti-angiogenic effects, inhibiting disease progression [15,16]. It has also been shown that the high expression of NTN4 is beneficial in improving the survival rate of breast cancer patients [17]. In conclusion, Netrins function differently in different types of cancer to promote or slow tumor progression. Therefore, Netrins may have a dual role in regulating the functions of tumors.

Bioinformatics techniques have been widely applied to investigate tumorigenesis and progression, with the availability of large amounts of biological data on the internet. However, the role of Netrins in ccRCC remains largely unknown. This study aimed to elucidate the potential prognostic value and therapeutic opportunities of the Netrin family in ccRCC patients by using bioinformatics technology as a guide to conduct experiments. We evaluated the expression profile and prognostic significance of Netrin genes in ccRCC using publicly available databases such as The Cancer Genome Atlas (TCGA), Cbioportal, and the Clinical Proteomic Tumor Analysis Consortium (CPTAC). In addition, the potential biological roles of the Netrin member NTN4 were further investigated and validated in ex vivo trials in ccRCC lines.

## 2. Materials and Methods

### 2.1. Data Collection

Gene expression data for 539 ccRCC cases were obtained from the TCGA database, along with clinical information for these patients. Additionally, 72 control cases were also included in the analysis. We also retrieved the GSE6344 [18] dataset from the GEO database, which included ten samples of ccRCC tissue and their corresponding normal kidney tissue samples. The microarray dataset utilized in the GSE6344 study was based on the GPL96 [HG-U133A] platform. The raw data were processed for normalization and correction using the “Limma” package in R software [19].

### 2.2. mRNA Expression Analysis

Initially, the expression levels of Netrin family members were evaluated in 27 tissue samples from the HPA-RNA project of the Stockholm Life Sciences Laboratory and an additional 27 tissue samples obtained from the GTEx database (https://www.ncbi.nlm.nih.gov/gene/59277/?report=expression, accessed on 8 March 2023). Subsequently, we compared the expression levels of the Netrin family genes in normal and cancerous kidney tissues from ccRCC patients (523 tumor samples and 72 control samples) and paired tumor samples (72 tumor samples and 72 matched control samples), using Wilcoxon rank-sum and paired t-tests to determine statistical significance, respectively.

### 2.3. Diagnostic and Prognostic Value Analysis

The operating characteristic curve (ROC) curve model of normal kidney tissue and ccRCC was constructed using the R language package “pROC.” [20]. The area can summarize the diagnostic performance under ROC.

We used the R language package “Survival” to conduct survival analysis, constructing univariate and multivariate Cox proportional hazard models. These models incorporated clinically relevant parameters in the evaluation of Netrin family members for overall survival (OS) and disease-free survival (DFS) in patients with kidney cancer. Additionally, the overall survival of ccRCC patients in the high- and low-expression groups was compared using Kaplan–Meier curves.

### 2.4. Tissue Specimens and Tumor Cells

Tumor and paracancerous specimens were collected from surgical patients at Renmin Hospital of Wuhan University between May 2020 and September 2022 after obtaining approval from the hospital’s Ethics Committee. The ccRCC cell lines (786-O, 769-P) utilized in our experiments were provided by Procell (PLST, Wuhan, China). The cells were cultured in a 37 °C incubator with 5% CO_2_ in RPMI-1640 medium supplemented with 10% FBS and 1% antibiotics (penicillin and streptomycin).

### 2.5. Western Blotting

Samples were treated with RIPA lysis buffer on ice for 30 min, with the addition of a protease inhibitor. The samples were subsequently stored at a temperature of −80 °C. An intermediate amount of protein from each sample was separated using 12% SDS polyacrylamide gels and then transferred to PVDF membranes (Millipore, NJ, USA). To account for differences in sample size, β-actin was used as an internal reference.

The detection of β-catenin requires the extraction of cellular nuclear proteins for analysis. Firstly, after washing with PBS, the precipitate was collected via centrifugation. To extract the cellular nuclear proteins, 20 µL of cell precipitate was suspended and dispersed by vortexing vigorously at the highest speed for 5 s in 200 µL of Cell Plasma Protein Extraction Reagent A (Beyotime, Shanghai, China). After ice bath, 10 µL of Cell Pulp Protein Extraction Reagent B (Beyotime, Shanghai, China) was added to the mixture, which was then centrifuged at 15,000× *g* for 5 min at 4 °C. The residual supernatant was completely aspirated and replaced with 50 µL of PMSF-added Nucleoprotein Extraction Reagent (Beyotime, Shanghai, China). The cell precipitate was again suspended and dispersed by vigorous vortexing for 15 s at high speed several times, and the mixture was finally centrifuged at 15,000× *g* for 10 min at 4 °C. The supernatant containing the extracted nucleoprotein was aspirated into a pre-cooled centrifuge tube. During electrophoresis, Lamin B was used as necessary as an internal reference for nuclear proteins.

The antibodies used in this study are listed in Table 1. Appendix A contains images of blots and molecular weight markers.

### 2.6. Cell Transfection

Lentiviral plasmids (vectors) for overexpression of NTN4 and control groups were synthesized by OBio Biotechnology (Shanghai, China). Cells derived from ccRCC were cultured in 6-well plates (Servicebio, Wuhan, China) and transduced with lentivirus-containing polybrene (8 g/mL). Following 24 h of cellular infection, the initial medium was replaced with a supplement containing 3 µg/mL puromycin, allowing for the selection of successfully constructed cell lines to be utilized in subsequent investigations.

### 2.7. CCK-8 Proliferation Assay

Cells that had been transfected were cultivated in 96-well plates, with each well containing 3000 cells. To evaluate cellular proliferation, 10 µL of CCK-8 solution (Beyotime, Shanghai, China) was added to each well, followed by incubation in a 96-well plate for 2 h. The optical density of the samples was then measured at 450 nm using a microplate reader.

### 2.8. Colony Growth Assay

Uniformly, 500 cells were seeded into a 6-well plate (Servicebio, Wuhan, China) and allowed to grow for 14 days. After fixation with 4% paraformaldehyde, the colonies were stained with gentian violet and rinsed with phosphate-buffered saline (PBS) (Servicebio, Wuhan, China). The number of visible colonies was then quantified to compare the colony-forming potential between experimental groups.

### 2.9. Transwell Experiment

Cell invasion capacity was evaluated using Transwell chambers with 8 μm pore size (Servicebio, Wuhan, China). The bottom of the migration chamber was initially coated with Matrigel (37 °C; incubated for 3 h). Cultured in a Transwell system, 200 µL of the serum-free medium was added to the upper chamber, while 800 µL of medium supplemented with 20% FBS was added to the lower chamber. After 24 h of incubation, the cells were washed with PBS, fixed in 4% PFA, and stained with gentian violet. Following air-drying, the cells were photographed utilizing a microscope and counted.

### 2.10. The Scratch Wound Assay

The cells were cultured until they reached 85–90% confluence in six-well plates. Subsequently, a 200-microliter pipette tip was used to create scratches in the cell monolayer of six-well plates. The scratches were washed three times and replaced with 1% FBS medium before photographing. The cells were further cultured in the medium, and images were obtained using a microscope.

### 2.11. Immunohistochemistry

The tissue sections were subjected to dewaxing and dehydration, followed by boiling in an appropriate buffer to retrieve the antigen. After retrieving the antigen, 3% hydrogen peroxide solution was used to block endogenous peroxidase activity in the sections. The sections were incubated with the primary antibody for over 36 h at 4 °C, followed by washing and incubation with a chromogenic-conjugated secondary antibody. Hematoxylin was used as the counterstain for the sections.

### 2.12. Immunofluorescence

Following successful transfection, cells were uniformly seeded onto slides and fixed with 4% paraformaldehyde solution before being permeabilized with 0.01% Triton X-100. Primary antibodies (Ki67, E-Cad, Vimentin, and β-catenin) were applied to the cells and allowed to incubate overnight at 4 °C. The cells were subsequently treated with fluorescently labeled secondary antibodies, and nuclear staining was performed with DAPI as the final step. Inverted fluorescence microscopy was used to observe fluorescence changes of Ki67, E-Cad, and Vimentin, while laser confocal microscopy was used to measure β-catenin’s fluorescence intensity and distribution within the cells.

### 2.13. Flow Cytometric Analysis

Cells were collected at 80% confluency, washed with PBS, and centrifuged. Following fixation in ethanol, the cells were stained with propidium iodide (PI) for 30 min, and their fluorescence signals were detected using the Beckman flow cytometry assay to determine the distribution of cell cycle phases. FlowJo software was used for data analysis and quantitative measurements. The horizontal coordinate reflected DNA content, while the vertical coordinate represented the number of cells. Percentages of each cell cycle phase and other relevant parameters were calculated to compare cell cycle changes between different groups.

Apoptosis rates were measured by flow cytometry, and cells were collected and fixed in the same way as the cycle assay. Cells were fixed and stained with Annexin V-FITC and PI for 15 min. Based on the binding of Annexin V-FITC and PI, the scatter plot graphed the relationship between fluorescent signal intensity and cell number. The quadrants were divided into lower left quadrant (LL) for normal cells, lower right quadrant (LR) for early apoptotic cells, upper right quadrant (UR) for late apoptotic cells, and left upper quadrant (UL) for necrotic cells. The proportion of cells at each stage was calculated to analyze the effects of apoptosis in different groups.

### 2.14. In Vivo Experiments

To establish an in vivo model, 15 sex-matched 4-week-old BALB/c mice were obtained from Hunan Slike Jingda Laboratory Animals (Changsha, China) and randomly assigned to 2 groups: Vector and NTN4. The mice were housed in a specific pathogen-free (SPF) environment at Renmin Hospital, Wuhan University experimental animal facility.

Eight nude mice were selected to establish a subcutaneous tumor model in the first group. Tumor cells from both groups were injected subcutaneously, and tumor growth was monitored every five days for one month. The mice were then euthanized, and the tumors were dissected, imaged, and weighed. Six nude mice were selected in the second group to establish a lung metastasis model. Tumor cells from two groups were injected via the tail vein. Two weeks later, the mice received an intraperitoneal injection of 15 mg/mL luciferase and were euthanized 15 min thereafter. The lung tissues were removed and placed on the darkroom imaging platform IVIS Lumina III, which reflected the degree and distribution of metastasis through luminescence of the lung. Finally, the tumor specimens were conserved for further experimentation by storing them in liquid nitrogen or formaldehyde.

### 2.15. Statistical Analysis

The data were analyzed using SPSS 22.0, GraphPad Prism 8.0, and other software. Statistical tests such as t and chi-square were used to analyze the data. The mean ± SD is used to express the results, unless specified otherwise. All experiments were repeated at least three times. The significance level was as follows: *** for *p* < 0.001, ** for *p* < 0.01, and * for *p* < 0.05.

## 3. Results

### 3.1. Expression of the Netrin Family in Normal Human Tissues

The authors explored the tissue specificity of Netrin family gene expression by analyzing 27 human tissue samples from the HPA-RNA project of Stockholm Life Sciences Laboratory and 27 human tissue samples from the GTEx database. Figure 1A,B shows the expression of Netrins in both databases. The expression of NTN1/3/4 in the two datasets is relatively similar. The mRNA expression level of NTN1 was highest in cardiac tissues, with the esophagus showing the next highest level. NTN3 had low mRNA expression in human tissues, mainly in the testes, liver, and endometrium, and NTN4 had wide mRNA distribution in human tissues, with high expression in the spleen and kidney. There were differences in the expression of other family members in the two datasets. The expression of NTNG1 was high in the retina and kidney in GTEx, and in the brain and kidney in HPA-RNA. Similarly, the expression of NTNG2 was high in the retina and prostate tissue in GTEx, and in the bone marrow and brain tissue in HPA-RNA.

As shown in Figure 1C, we further investigated the protein expression of Netrin in human tissues by analyzing immunohistochemistry data from 44 different tissues in the HPA project. The results showed that NTN1 was only expressed in testis tissue; NTN5/G1/G2 also had trace expression in very few tissues, but NTN3 and NTN4 had widespread expression in human tissues containing cavities, such as NTN3 in the esophagus, bronchus, and nasal cavity as well as kidneys. NTN4 had high expression in gastrointestinal tract and urogenital system tissues.

### 3.2. Expression of the Netrin Family in ccRCC

We evaluated the expression of each member of the Netrin family in ccRCC, using the TCGA database. The expression of NTN5 and NTNG2, among the six Netrin family members, was significantly upregulated in ccRCC tissues; NTN3 was downregulated in ccRCC samples, but no significant difference was observed in 72 paired samples included in the TCGA database; the other three genes (NTN1, NTN4, and NTNG1) were significantly downregulated in ccRCC tissues (Figure 2A,B). To further assess the accuracy of Netrin family members in distinguishing ccRCC tissues from normal tissues, we constructed ROC models using the TCGA database. The data are presented in Figure 2C,D. Except for NTN3, the expression levels of NTN1/4/5, NTNG1, and NTNG2 had good diagnostic value for ccRCC (AUC > 0.8).

### 3.3. Correlation of Netrin Family Members with Survival Prognosis in Patients with ccRCC

We conducted a single-gene Cox survival analysis of OS and DFS for Netrin family member genes in ccRCC patients using TCGA (Figure 3A,B). The results of OS single-gene Cox analysis indicated that only NTN4 was a protective factor in ccRCC (HR < 1), while NTN5, NTNG1, and NTNG2 were all risk factors (HR > 1). The results of DFS single-gene Cox analysis indicated that NTN4 was a protective factor in ccRCC (HR < 1), while NTN3, NTNG1, and NTNG2 were risk factors (HR > 1). Next, we performed a Kaplan–Meier survival analysis of ccRCC using the TCGA (Figure 3C,D). Our analysis revealed that low expression of NTN1 was associated with better OS prognosis in ccRCC, and low expression of NTN4 was associated with better prognosis in both OS and DFS. By contrast, patients in the high-expression groups of NTNG1/G2 showed better OS and DFS prognosis than those in the low-expression groups.

### 3.4. Independent Diagnostic Value of Netrin Family Members in the Survival of Patients with ccRCC

To evaluate the independent prognostic value of NTN4, NTNG1, and NTNG2 genes for OS and DFS in ccRCC patients, Cox analysis was performed (Figure 3E). Univariate Cox analysis indicated that high NTN4 expression was associated with better prognosis for both OS and DFS, while low expression of NTNG1/G2 was associated with better prognosis for both OS and DFS. The HR of NTN4 in the multivariate Cox analysis model of ccRCC was less than 1, indicating its independent protective role in ccRCC. Conversely, the HR of NTNG1/G2 in the multivariate Cox analysis model of ccRCC was less than 1, indicating that NTNG1/G2 were independent risk factors in ccRCC. Additional analysis of the data combined with the clinicopathological characteristics revealed that in patients with NTN4 low expression of ccRCC, the death rate was higher if the corresponding patient was older or had a higher-grade stage pathology. Likewise, in patients with high NTNG1/G2 expression, the morbidity and mortality rates were higher if the grading stage of the patient’s pathology was advanced.

### 3.5. Expression of Netrins Family Members in Patients with ccRCC

We further validated NTN4, NTNG1/G2 expression using GSE6344 and CPTAC datasets related to ccRCC, but only NTN4 showed differential expression in both databases. Figure 4A,B shows the results. NTN4 mRNA expression was downregulated in RCC samples from GSE6344 and protein expression was also downregulated in ccRCC samples from CPTAC. We also used immunohistochemical data from the HPA to study NTN4 protein expression distribution in ccRCC and normal samples, as depicted in Figure 4C. The expression level of NTN4 was lower in ccRCC patients than in healthy individuals. Finally, we conducted Western blot and immunohistochemical experiments to verify NTN4 expression in ccRCC using six paired tissue specimens from ccRCC and normal kidney tissues obtained at the Renmin Hospital of Wuhan University. The results are shown in Figure 4D,E, demonstrating that NTN4 expression was decreased in RCC compared with normal tissues, consistent with the findings from the online database.

### 3.6. Overexpression of NTN4 Inhibits the Growth of ccRCC Cells In Vitro

The effect of NTN4 overexpression on the biological behavior of 786-O and 769-P ccRCC cells in vitro was investigated. We initially assessed NTN4 expression in commonly used ccRCC cell lines with HK2 cells as the control. Figure 5A shows that NTN4 expression was relatively low in 786-O and 769-P. We then established cells stably expressing the NTN4 gene in these two cell lines. As Figure 5B shows, Western blotting confirmed that NTN4 expression was significantly higher in the pc-NTN4 group than in the pc-Vector group.

We assessed cell viability using the CCK-8 assay and found that NTN4 overexpression reduced the metabolic activity of ccRCC cells from day 3 compared with the pc-Vector group (*p* < 0.01) (Figure 5C), indicating that NTN4 slowed ccRCC cell growth. We also performed a colony formation assay for 2 weeks and found that NTN4 overexpression reduced the clonogenic ability of both 786-O (*p* < 0.05) and 769-P (*p* < 0.01) cells (Figure 5D). Moreover, immunofluorescence detection of Ki-67 revealed reduced expression in the pc-NTN4 group, indicating decreased cell proliferation ability compared with the pc-Vector group (Figure 5E).

### 3.7. We Evaluated the Effect of NTN4 Overexpression on ccRCC Cell Migration In Vitro

We conducted a scratch wound assay and observed a significantly smaller wound area in the pc-NTN4 group compared with the pc-Vector group after 24 h in 786-O and 769-P cells (*p* < 0.001), as shown in Figure 6A,B. We also performed a transwell invasion assay and found that the pc-NTN4 group had fewer migrating cells than the pc-Vector group, indicating reduced invasion ability (Figure 6C,D). Using fluorescence microscopy, we investigated the association between NTN4 and the process of epithelial–mesenchymal transition (EMT). By double fluorescent labeling (Figure 6E), we observed increased expression of the epithelial marker E-cadherin and decreased expression of the mesenchymal marker vimentin in the pc-NTN4 group. Expression levels of N-cadherin, matrix metalloproteinase 2 (MMP-2), and Vimentin in the pc-NTN4 group were reduced, which was confirmed by Western blotting (Figure 6F). NTN4 inhibited ccRCC cell proliferation, migration, and invasion.

### 3.8. Regulation of the Cell Cycle and Apoptosis by NTN4

We performed PI staining in flow cytometry to evaluate the cell cycle distribution of 786-O and 769-P cell lines. The pc-NTN4 group exhibited a higher percentage of cells in the G1 phase and more apoptotic cells compared with the pc-Vector group, as shown by flow cytometry (Figure 7A–D) and TUNEL staining (Figure 7E) in both 786-O and 769-P cells, suggesting cell cycle arrest and increased apoptosis. Western blotting analysis revealed that the pc-NTN4 group exhibited decreased cyclin D1 expression, increased expression of the pro-apoptotic proteins Bax and cleaved caspase-3, and decreased expression of the anti-apoptotic protein Bcl-2 (Figure 7F). In summary, NTN4 overexpression induced ccRCC cell cycle arrest and apoptosis.

### 3.9. NTN4-Mediated Regulation of the WNT/β-Catenin Signaling Pathway in ccRCC

We explored how NTN4 regulates ccRCC tumorigenesis and metastasis, using KEGG enrichment analysis. NTN4 was associated with several oncogenic pathways, including the Wnt/β-catenin signaling pathway (Figure 8A). Using Western blotting, the effect of NTN4 on β-catenin signaling was assessed. A decrease in the expression and nuclear translocation of β-catenin was observed in 786-O and 769-P cells transfected with pc-NTN4 compared with those transfected with pc-Vector (Figure 8B). Nuclear translocation of β-catenin is a hallmark of the Wnt pathway. The NTN4 high-expression group exhibited reduced β-catenin fluorescent particles in the cytoplasm and nucleus, indicating inhibition of both β-catenin expression and translocation to the nucleus, as observed through laser confocal microscopy (Figure 8C). We also analyzed GSK3β, which plays a role in the degradation complex of β-catenin and regulates its nuclear translocation and signaling. We found that NTN4 overexpression inhibited GSK3β phosphorylation (Figure 8B). In summary, NTN4 inhibited Wnt/β-catenin signaling via GSK3β in ccRCC cells.

### 3.10. Enhancement of β-Catenin Signaling Can Counteract the Impact of NTN4

We investigated whether NTN4 overexpression affects Wnt/β-catenin signaling in ccRCC cells. Using Western blotting, we evaluated the expression levels of cyclin D1 and matrix metalloproteinases (MMPs), which are direct targets of Wnt/β-catenin signaling. NTN4 overexpression downregulated cyclin D1 and MMP2 expression in ccRCC cells (Figure 6 and Figure 7F). To confirm this observation, we treated ccRCC cells with Laduviglusib trihydrochloride (CHIR-99021), a potent activator of the Wnt/β-catenin signaling pathway. We hypothesized that CHIR-99021 would inhibit the NTN4-mediated cellular phenotype.

We evaluated the impact of CHIR-99021 on the growth, movement, and infiltration of ccRCC cells through CCK8, wound healing, and Transwell assays, respectively. We observed that CHIR-99021 partially alleviated the inhibitory impact of NTN4 overexpression on these cellular functions (Figure 8D and Figure 9A–D). We also assessed the impact of CHIR-99021 on EMT markers, regulators of the cell cycle, and apoptosis-related proteins using Western blotting. We found that CHIR-99021 partially restored EMT marker, cyclin D1, and antagonized changes in Bax, cleaved caspase-3, and Bcl-2 expression induced by NTN4 overexpression (Figure 9E). Our findings indicate that NTN4 inhibits the activity of the Wnt/β-catenin signaling pathway in ccRCC cells.

### 3.11. Subcutaneous Proliferation Model and Lung Metastasis Model in Nude Mice

We next evaluated the antitumor effects of NTN4 overexpression in vivo using two mouse models of ccRCC. We subcutaneously injected 786-O cells transfected with pc-Vector or pc-NTN4 into nude mice and observed tumor growth for four weeks. The group of mice injected with pc-NTN4 transfected cells showed a statistically significant reduction in tumor size and weight compared with the control group (Figure 10A,B). Immunohistochemical staining revealed reduced Ki67 expression in the NTN4 group, indicating decreased cell proliferation (Figure 10C). Western blotting analysis confirmed that NTN4 overexpression downregulated β-catenin, Vimentin, and Bcl-2 expression in the tumor tissues (Figure 10D).

Second, we intravenously injected fluorescently labeled 786-O cells into BALB/c nude mice and monitored lung metastasis by in vivo imaging. Two weeks later, we observed a reduction in lung metastases and tumor invasion in the NTN4 group compared with the control group, as evidenced by fewer metastatic nodules (Figure 10E) and decreased tumor cell infiltration in lung tissues, as shown by hematoxylin and eosin staining (Figure 10F). The findings suggest that overexpression of NTN4 suppresses the migratory and invasive properties of ccRCC cells in vivo.

In conclusion, our findings demonstrate that NTN4 functions as a tumorigenesis suppressor gene, restraining the proliferation and metastasis of ccRCC cells in vitro and in vivo.

## 4. Discussion

Netrins are known to act as important regulators of cell migration and axon guidance during embryogenesis [21]. However, the roles of different genes within the Netrin family can be paradoxical, with NTN1 being upregulated in colorectal cancer-associated fibroblast cells and promoting cancer cell stemness [22,23]. In addition, NTN1 has been demonstrated to enhance the progression of colorectal cancer by releasing anti-apoptotic signals through the NF-kappaB pathway in individuals with inflammatory bowel disease [24]. Conversely, NTN4 is a tumor growth inhibitor in primary and metastatic colorectal tumors [25]. Studies have indicated that NTN4, acting as a susceptibility locus, is capable of inhibiting breast cancer growth and reducing the invasiveness of cancer cells through the regulation of EMT [26,27]. Despite these studies, the Netrin family’s expression and potential prognostic impact in ccRCC remain unclear. Therefore, we sought to investigate the Netrin family’s expression and diagnostic and prognostic value in ccRCC and to elucidate their potential function in this disease.

We initially analyzed the mRNA expression of the Netrin family in two databases and its protein expression in the Human Protein Atlas Project [28,29,30]. Our findings revealed similarities in the expression of NTN3/4. In healthy human tissues, NTN3 was detected in various amounts in the skin, esophagus, bronchus, nasal cavity, kidney, and vagina. In contrast, NTN4 exhibited high expression levels in the skin, digestive system, and certain organs of the genitourinary system. Both were detected in organs with luminal tracts, suggesting that their function is related to the extracellular matrix protein laminin. Therefore, they may play a role in specific organs, potentially affecting tumor growth and migration in those locations [31,32].

NTN1/5 and NTNG1/G2 mRNA expression are also characteristic of the Netrin family, with high concentrations in the nervous system or tissues with secretory functions. However, their activity may become dormant following their involvement in early nervous system development or due to posttranscriptional regulatory mechanisms that prevent their expression in tandem with other proteins [33]. For example, NTN1 is normally detected in small amounts in the testis. However, in rectal tumors or inflammatory diseases, NTN1 expression is restored, leading to disease progression. The differing expression profiles of NTN1 and NTN3/4 suggest that they may have opposing roles in tumors.

The precise prognostic and biological implications of the Netrin family in ccRCC remain largely unknown. To address this, we examined various Netrin family members’ expression patterns and clinical relevance in a cohort of ccRCC patients. Through our differential analysis, we observed significant dysregulation of NTN1/3/4/G1, downregulated, and NTN5/G2, upregulated, in ccRCC tissues compared with normal renal tissues in the TCGA dataset. Despite the observed differences in expression, all Netrin family members examined had strong diagnostic potential for identifying ccRCC, as evidenced by the robust performance of the ROC model. These findings suggest that Netrin family members could be promising biomarkers for ccRCC diagnosis.

We examined the correlation between Netrin family members and the prognostic outcomes of ccRCC patients. Low expression of NTN4 was found to be an independent protective factor for both OS and PFS in ccRCC patients, as our results demonstrated. In contrast, decreased expression of NTNG1/G2 was found to be an independent risk factor for poor survival prognosis in both overall survival (OS) and progression-free survival (PFS) among ccRCC patients. Our findings suggest that NTN4 may have oncogenic potential, while NTNG1/G2 may have a pro-oncogenic role in the development of ccRCC.

Furthermore, we validated our results by analyzing the independent datasets GSE6344 and CPTAC, showing that only NTN4 was differentially expressed in both datasets [34]. Additionally, immunohistochemical analysis of ccRCC specimens included in the HPA confirmed that NTN4 was significantly downregulated in ccRCC [35], which was also replicated in clinical specimens from the Renmin Hospital of Wuhan University. Our findings suggest that NTN4 could be a promising clinical therapeutic target for ccRCC.

NTN4 is a prognostic biomarker in ccRCC, which corroborates the findings of Yi et al., who reported that NTN4 is also a prognostic biomarker in breast cancer [36]. Furthermore, NTN4 inhibits tumor progression in colorectal cancer by exerting anti-angiogenic effects [16]. Therefore, elucidating the role of NTN4 in ccRCC is our next research aim.

We investigated the effect of NTN4 on the growth, invasion, and apoptosis of ccRCC cells, by restoring their expression in 786-O and 769-P cell lines. Cell viability assays and other assays revealed that NTN4 suppressed ccRCC cell proliferation. Regulating the transition from G1 to S phase in the cell cycle is a crucial function of Cyclin D1 [37]. We observed that NTN4 induced downregulation of cyclin D1 and increased G1 phase arrest. Flow cytometry and Tunel staining for apoptosis also indicated that overexpression of NTN4 enhanced the apoptosis rate. The proteins Bax, Bcl-2, and caspase-3 are all part of the complex regulatory network that governs the process of apoptosis [38], and NTN4 also modulated their expression and activation, resulting in increased apoptosis. These results suggest NTN4 impedes the normal cell cycle and proliferation process and induces apoptosis.

EMT facilitates tumor cell dissemination and distant organ colonization, critical steps in tumor recurrence and metastasis. E-cadherin and mesenchymal markers are indicators of EMT [39]. Moreover, matrix metalloproteinase-2 (MMP-2) also participates in the EMT process [40]. We demonstrated that NTN4 upregulated E-cadherin expression while downregulating N-cadherin, MMP-2, and Vimentin expression. Furthermore, EMT is implicated in cancer stem cell generation and therapy resistance, and targeting tumor EMT represents a potential therapeutic strategy [41]. Our established in vivo models of ccRCC proliferation and lung metastasis demonstrated NTN4’s inhibitory effects on tumor growth and metastasis. Therefore, overexpression of NTN4 attenuates tumor migration and invasion and provides a novel direction for ccRCC treatment.

Our study revealed that NTN4 exerts an inhibitory effect on the proliferation, migration, and invasion of ccRCC cells through the activation of the Wnt/β-catenin signaling pathway. Dysregulated Wnt signaling has been associated with tumor recurrence and metastasis in multiple types of cancer, including colorectal, breast, and renal malignancies [42,43,44]. Current research indicates that modulation of certain signaling molecules can impact the Wnt/β-catenin signaling cascade and alter the progression of cancer. For example, in breast cancer, PROX1 has been shown to stimulate the Wnt/β-catenin signaling pathway, thereby enhancing EMT and promoting distant metastasis [45]. In contrast, in ccRCC, ATF3 can inhibit the β-catenin signaling pathway, slowing distant metastasis [46]. Targeting the Wnt signaling pathway shows promise as a therapeutic approach in treating ccRCC. In vivo studies have indicated that inhibiting the Wnt pathway can impede the growth of ccRCC tumors and their cancer stem cell populations [47]. Our study demonstrated the inhibitory effect of NTN4 on the expression of proteins related to the Wnt/β-catenin pathway. Furthermore, we verified that activation of β-catenin could partly reverse NTN4’s suppressive impact on the proliferation and migration of ccRCC cells. In conclusion, NTN4 can act as a Wnt pathway inhibitor to restrain ccRCC progression, and represents a potential therapeutic opportunity for ccRCC.

## 5. Conclusions

Our study suggests that NTNG1/G2 and NTN4, members of the Netrin family, may have prognostic value as biomarkers for ccRCC patients. Specifically, NTNG1/G2 was identified as an independent risk factor and NTN4 as an independent protective factor in ccRCC, providing valuable insights into the prognostic potential of Netrin family members in this disease. Our research provides evidence that NTN4 functions as an oncosuppressor gene in ccRCC by impeding the growth and movement of cells and promoting cell death through modulation of the Wnt/β-catenin signaling pathway. Consequently, our findings indicate that investigating NTN4 has the potential to enhance diagnosis, treatment, and outcomes for ccRCC patients.

## Figures and Tables

**Figure 1 cancers-15-02816-f001:**
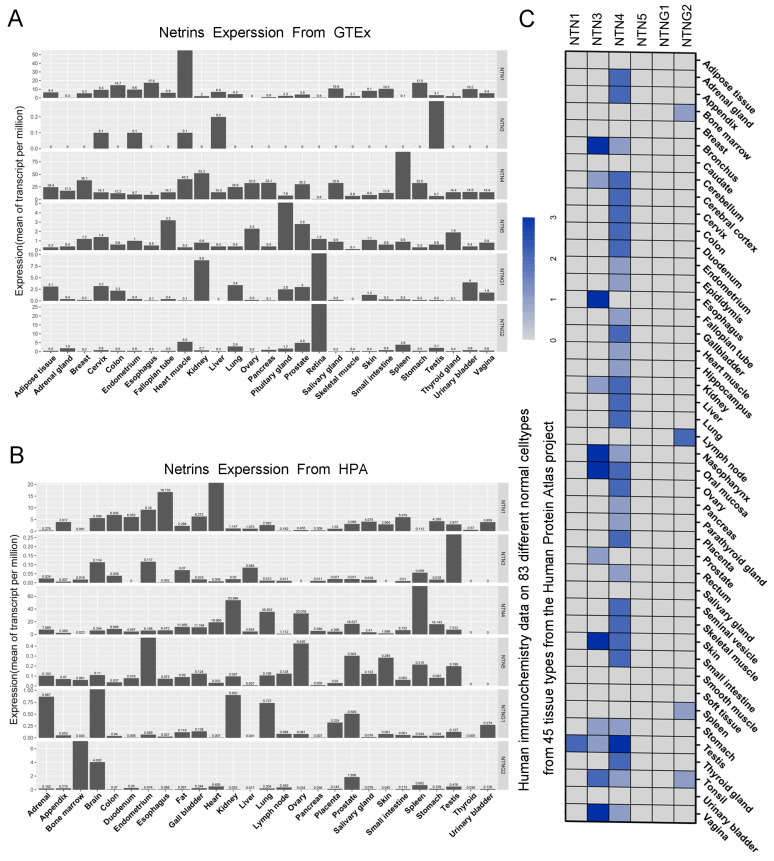
Expression characteristics of Netrin gene family members. (**A**) RNA expression distribution of Netrin genes in 27 tissue types. Depicted by using the transcriptomics dataset GTEx. (**B**) RNA expression distribution of Netrin genes in 27 tissue types. Described using the HPA-RNA project of the Stockholm Life Sciences Laboratory. (**C**) Netrins protein expression distribution in human tissues. Described based on human immunochemical staining data from 44 tissue types in the Human Protein Atlas Project.

**Figure 2 cancers-15-02816-f002:**
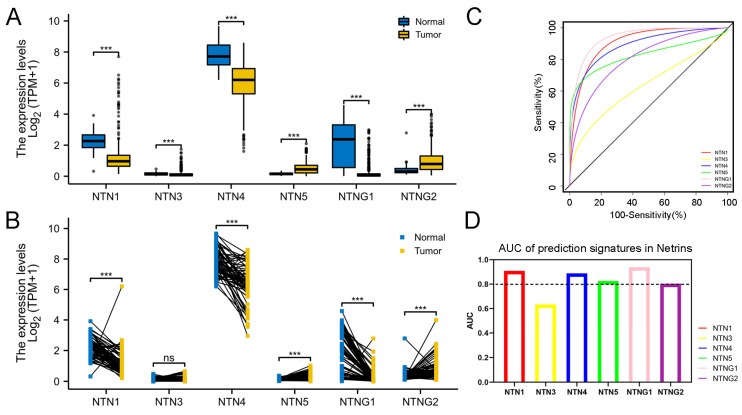
Expression of Netrin gene family members in ccRCC. (**A**) Expression of the Netrin family in ccRCC and normal tissues in the TCGA. In total, 539 cases of ccRCC tissues and 72 cases of normal tissues were included. (**B**) Evaluation of Netrin family expression in the TCGA across ccRCC and adjacent normal tissues. A total of 72 cases of ccRCC and paracancerous tissues. (**C**) The study utilized the TCGA database to construct ROC curves, with sensitivity plotted on the vertical axis and (100%—specificity) plotted on the horizontal axis. (**D**) The diagnostic accuracy of Netrin family members was evaluated using the area under the curve. (*** for *p* < 0.001, ns for no significant *p* value).

**Figure 3 cancers-15-02816-f003:**
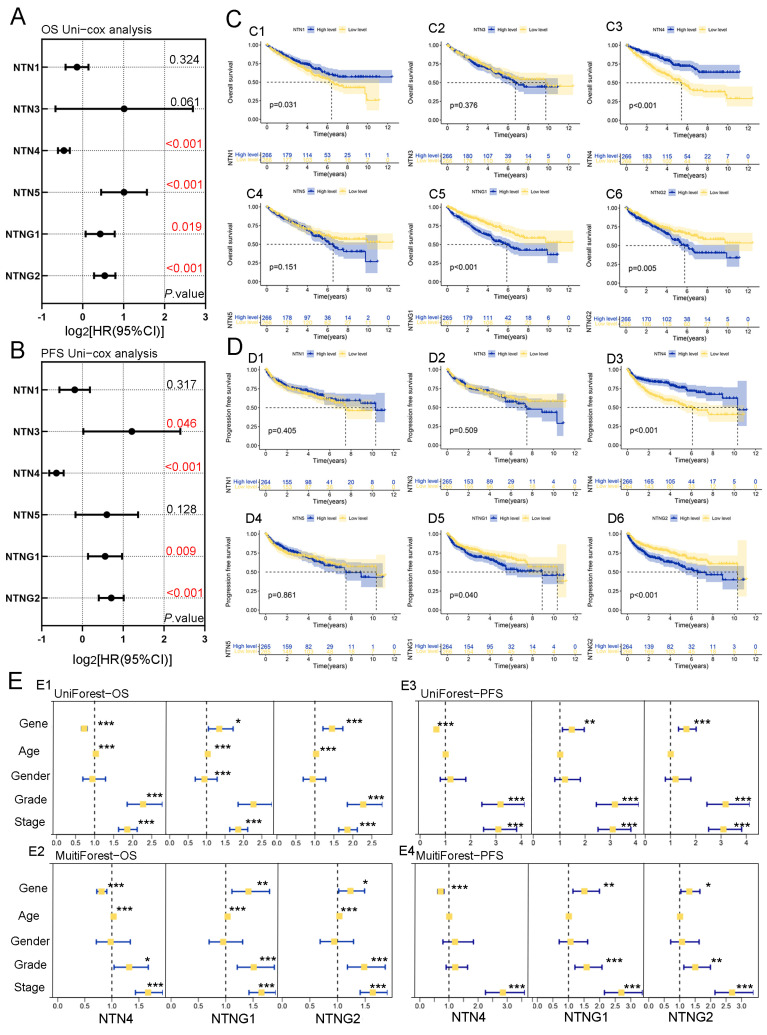
The prognostic value of the Netrin family gene in patients with ccRCC. (**A**) The expression of Netrins was analyzed using univariate Cox regression to investigate its association with OS in ccRCC. NTN4/5 and NTNG1/2 genes were associated with OS in ccRCC. The *p*-value for the Cox regression analysis of this gene in relation to OS is highlighted in red font, indicating its statistical significance. (**B**) The expression of Netrins was analyzed using univariate Cox regression to investigate its association with PFS in ccRCC. NTN3/4 and NTNG1/2 genes were associated with PFS in ccRCC. The *p*-value for the Cox regression analysis of this gene in relation to PFS is highlighted in red font, indicating its statistical significance. (**C**,**D**) OS/PFS survival analysis of Netrin genes in ccRCC. (**E**) Univariate and multivariate Cox regression analyses were also performed to investigate the potential prognostic value of NTN4 and NTNG1/G2 in ccRCC for OS/PFS. These genes could be considered as independent prognostic factors for ccRCC. *** for *p* < 0.001, ** for *p* < 0.01, and * for *p* < 0.05.

**Figure 4 cancers-15-02816-f004:**
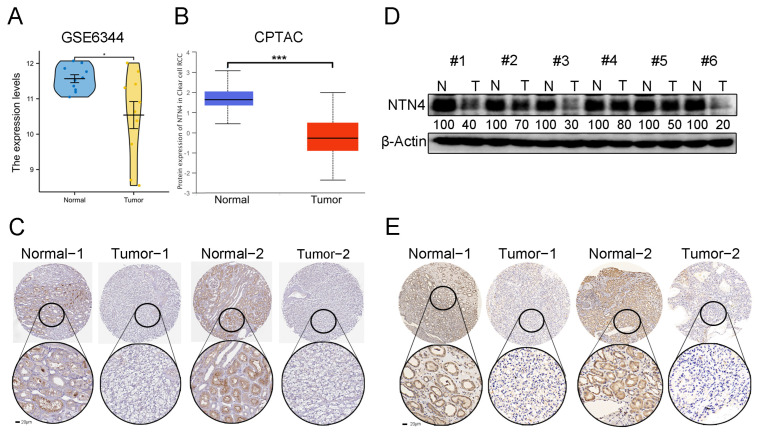
NTN4 expression in kidney tissues and ccRCC. (**A**) Validation of NTN4 expression in dataset GSE6344. (**B**) Validation of NTN4 expression at the CPTAC (http://ualcan.path.uab.edu/analysis-prot.html, accessed on 8 March 2023). (**C**) Two sets of representative images of NTN4 in ccRCC and kidney tissues included in the HPA database (www.proteinatlas.org/pathology, accessed date: 8 March 2023) (scale bar of 20 μm). (**D**) Levels of NTN4 protein measured in healthy kidney tissues and ccRCC tissues. (**E**) Protein levels of NTN4 in normal kidney and ccRCC tissues were analyzed by immunohistochemical assays (scale bar of 20 μm). *** for *p* < 0.001 and * for *p* < 0.05. The uncropped bolts are shown in Appendix A.

**Figure 5 cancers-15-02816-f005:**
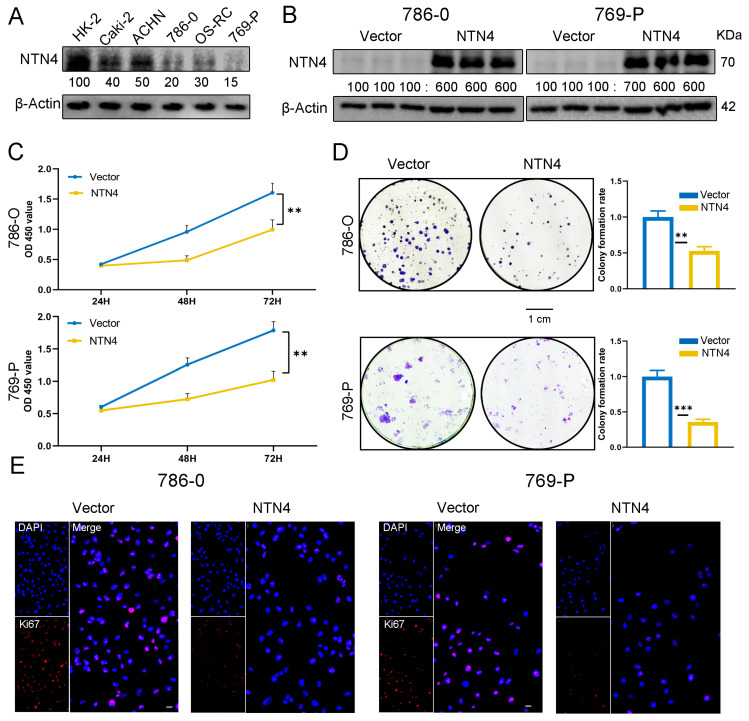
Overexpression of NTN4 inhibits the proliferation of ccRCC cells in vitro. (**A**) Expression of NTN4 in five ccRCC cell lines (Caki-2, OSRC2, 786-O and 769-P) and one control cell line (HK2) was analyzed by Western blotting. (**B**) Validation of the effect of NTN4 overexpression in 786-O and 769-P. Molecular weight units are kDa. The uncropped bolts are shown in Appendix A. (**C**) The proliferative capacity of the cells was compared using the CCK-8 kit. (**D**) Comparison of the number of cell clones in NTN4 overexpressed and control cells. (scale bar: 1 cm) (**E**) Immunofluorescence assay to detect the expression of Ki67 in cells. (Scale bar 50 um). *** for *p* < 0.001 and ** for *p* < 0.01.

**Figure 6 cancers-15-02816-f006:**
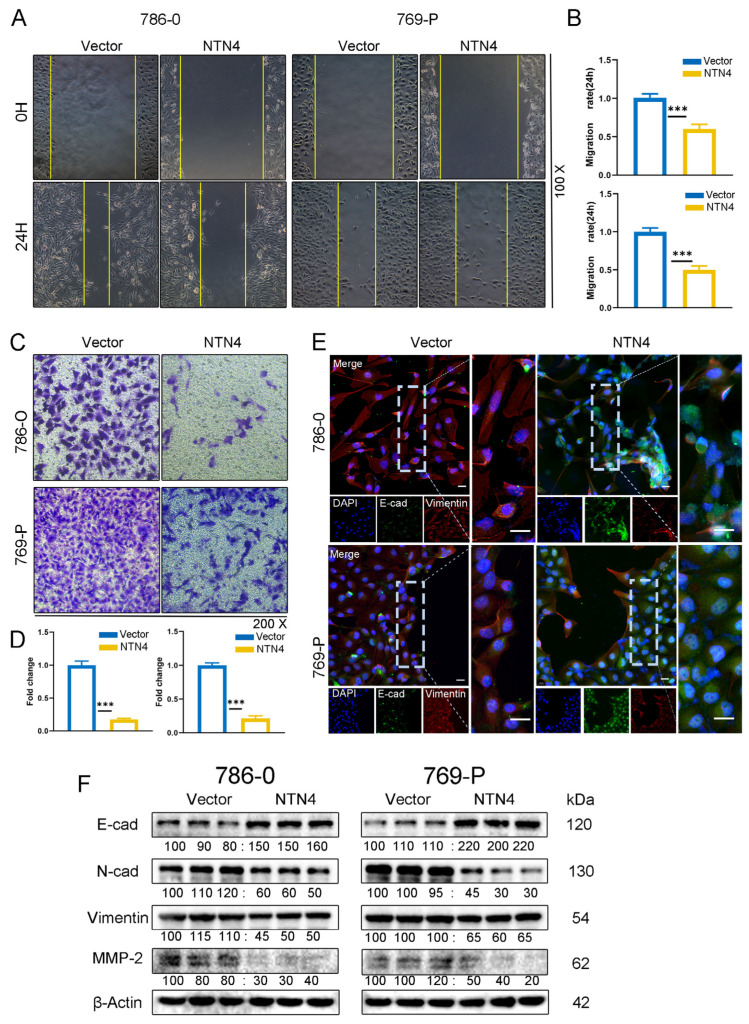
Overexpression of NTN4 suppressed both migration and invasion of ccRCC cells in vitro. (**A**,**B**) Scratch healing assay to assess the migration of ccRCC cells. (**C**,**D**) Cell invasion ability of both groups was assessed by Transwell invasion assay (with Matrigel gel). (**E**) Double immunofluorescence evaluation of E-cad and Vimentin expression in NTN4 overexpression group and control group. The dotted line serves as a guide to indicate the enlarged portion of the image. (scale bar: 20 μm) (**F**) Alteration of EMT-related markers in NTN4 overexpression group and control group were verified by Western blotting. *** for *p* < 0.001. The uncropped bolts are shown in Appendix A.

**Figure 7 cancers-15-02816-f007:**
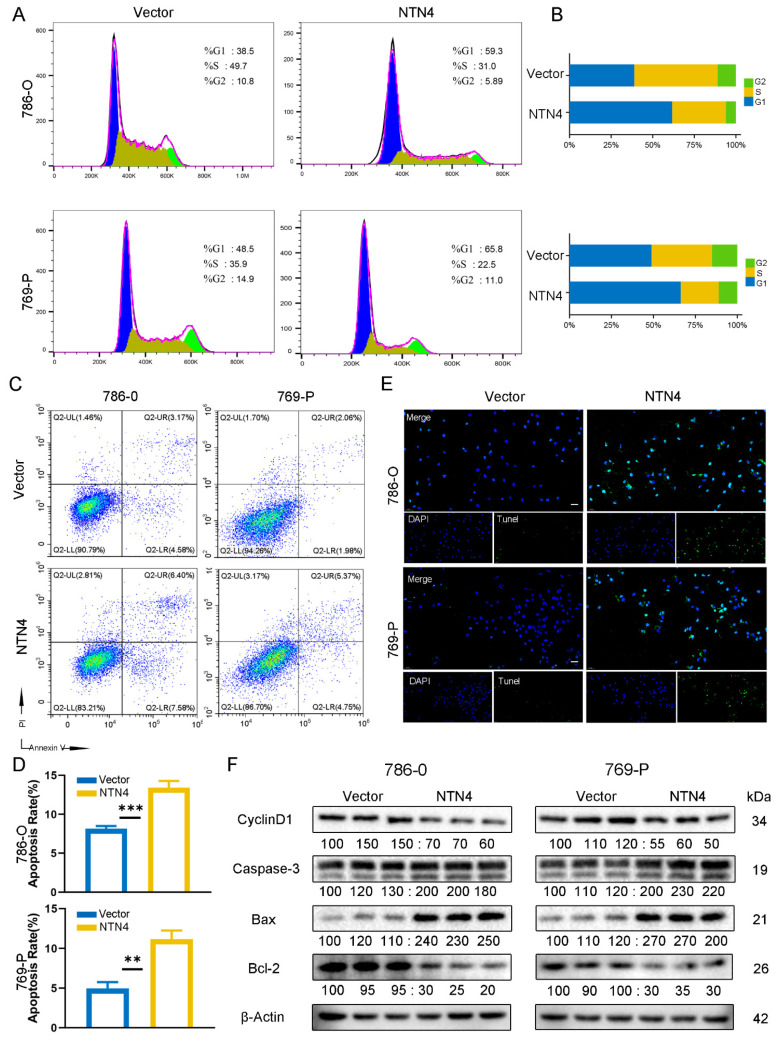
The overexpression of NTN4 in ccRCC cells was found to induce cell cycle arrest and apoptosis. (**A**) Detection of NTN4 overexpression and control 786-O and 769-P cell cycle alterations by flow cytometry. (**B**) Quantitative analysis of cell cycle changes. (**C**) Comparison of apoptosis rates of NTN4 overexpression and control cells by flow cytometry. (**D**) Quantitative analysis of apoptotic rate changes. (**E**) An immunofluorescence assay was conducted to compare TUNEL staining between NTN4 overexpression and control cells. (200X, scale bar: 50 μm). (**F**) After overexpression of NTN4 in cells, the expression of cell cycle protein D1 and apoptosis markers including Bax, cleaved caspase-3, and Bcl-2 were validated. *** for *p* < 0.001 and ** for *p* < 0.01. The uncropped bolts are shown in Appendix A.

**Figure 8 cancers-15-02816-f008:**
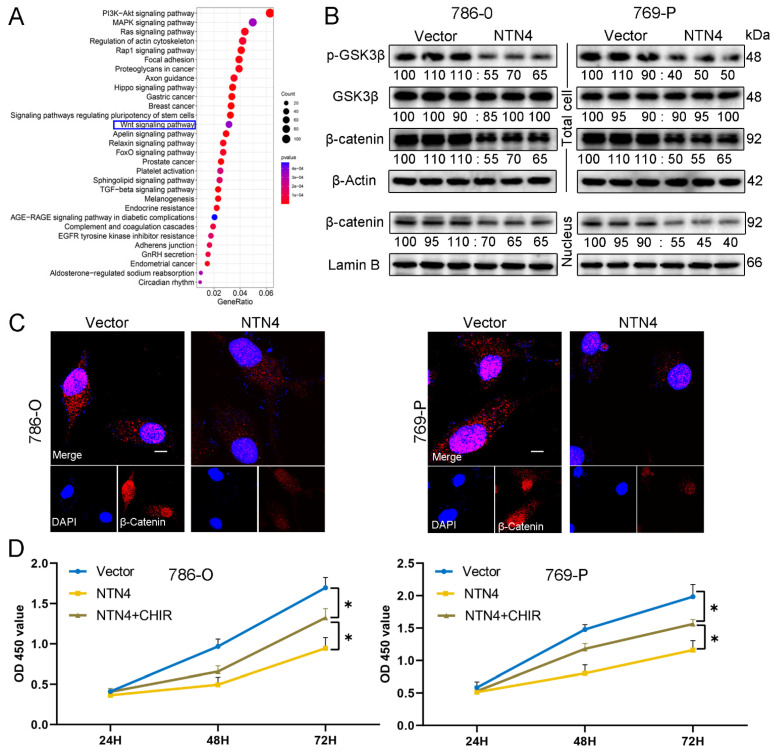
NTN4 affects ccRCC cell proliferation through the Wnt/β-catenin pathway. (**A**) Enrichment results of KEGG pathway; NTN4 may be involved in ccRCC progression through Wnt/β-catenin pathway. (**B**) Western blotting analysis of Wnt/β-catenin pathway proteins. The uncropped bolts are shown in Appendix A. (**C**) The expression and nuclear translocation of β-catenin in ccRCC cells were revealed by laser confocal microscopy (1000X; Scale bar: 10 μm). (**D**) After NTN4 overexpression, treatment with the β-catenin activator CHIR partially reversed the inhibition of cell proliferation capacity. * for *p* < 0.05.

**Figure 9 cancers-15-02816-f009:**
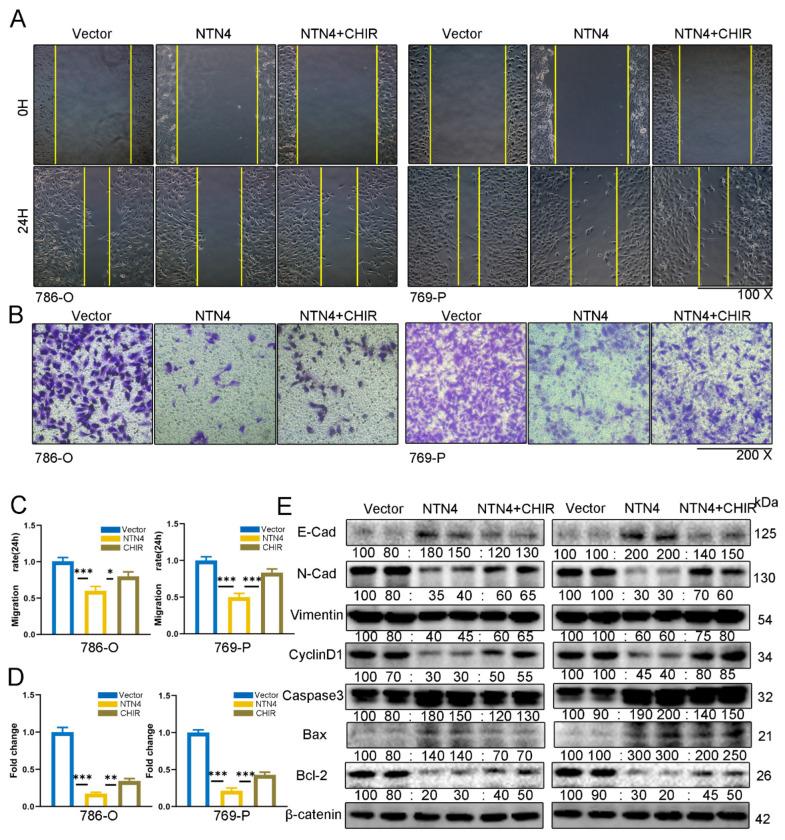
The Wnt/β-catenin pathway is implicated in the effect of NTN4 on ccRCC cell migration and invasion. (**A**,**B**) The ability of cells to migrate was partially restored by CHIR, an activator of β-catenin, in the presence of overexpressed NTN4; quantitative analysis of cell migration ability. (**C**,**D**) β-catenin activator CHIR restored the effect of NTN4 overexpression on cell migration and invasion ability; quantitative analysis of cell invasion ability. (**E**) Western blotting analysis of EMT, cell cycle and changes in key proteins key proteins during apoptosis. *** for *p* < 0.001, ** for *p* < 0.01, and * for *p* < 0.05. The uncropped bolts are shown in Appendix A.

**Figure 10 cancers-15-02816-f010:**
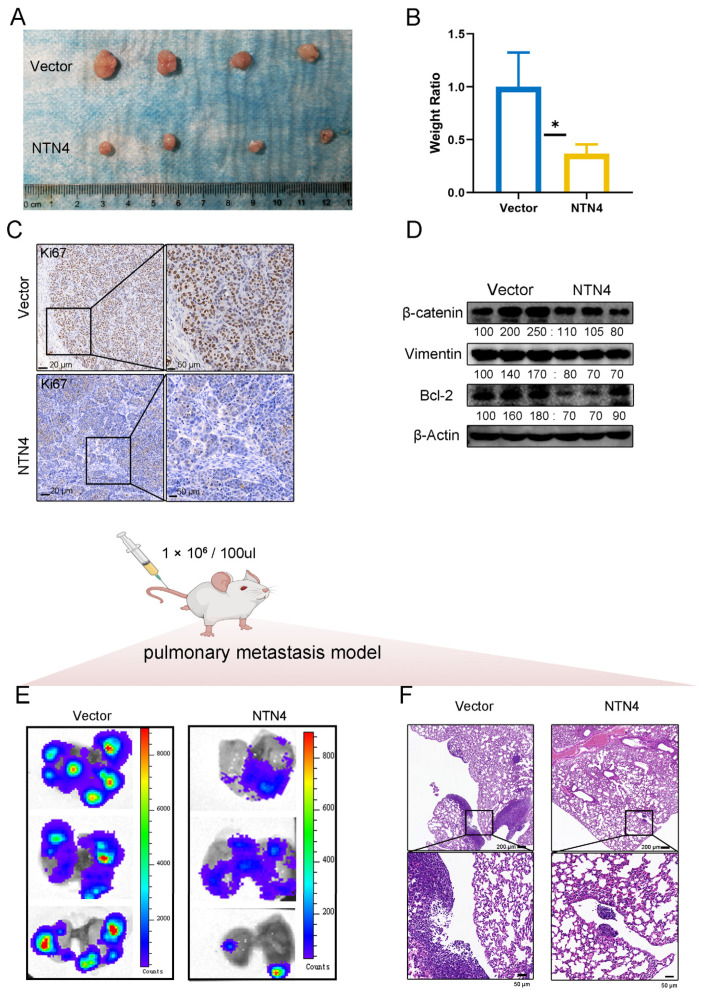
Upregulation of NTN4 expression inhibits the growth and metastasis of ccRCC cells in vivo. (**A**) Comparison of tumor model sizes in Vector, NTN4 groups. (**B**) Quantification of tumor weight in Vector, NTN4 groups. (**C**) Images of Ki67 staining in tissues detected by immunohistochemistry. (**D**) Changes of Vimentin, β-catenin, and Bcl-2 protein in tumors of Vector, NTN4 group were analyzed by Western blotting. The uncropped bolts are shown in Appendix A. (**E**) Fluorescence imaging by live imager to observe lung metastasis in Vector, NTN4. (**F**) HE-stained images of lungs in the Vector, NTN4 group. * for *p* < 0.05.

**Table 1 cancers-15-02816-t001:** Listed below are the primary antibodies used in this research.

Antibody	Specificity	WB	IF	IHC	Company
Bax	Rabbit	1:800	−	−	Abcam
Bcl-2	Rabbit	1:800	−	−	Abcam
Cleaved caspase-3	Rabbit	1:800	−	−	Abcam
Cyclin D1	Rabbit	1:800	−	−	Proteintech
E-cadherin	Rabbit	1:800	1:100	−	Abcam
GSK3β	Rabbit	1:800	−	−	Proteintech
Ki67	Rabbit	1:800	1:100	1:200	Proteintech
Lamin B	Rabbit	1:800	−	−	Proteintech
MMP-9	Rabbit	1:800	−	−	Proteintech
N-cadherin	Rabbit	1:800	1:100	−	Abcam
NTN4	Rabbit	1:500	−	1:200	ABclonal
p-GSK3β	Rabbit	1:800	−	−	Proteintech
Vimentin	Rabbit	1:800	1:100	−	Abcam
β-actin	Rabbit	−	−	−	Proteintech
β-catenin	Rabbit	1:800	1:100	−	Proteintech

## Data Availability

Data are contained within the article.

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
