# Peer review of "Netrin Family Genes as Prognostic Markers and Therapeutic Targets for Clear Cell Renal Cell Carcinoma: Netrin-4 Acts through the Wnt/β-Catenin Signaling Pathway"

_cancers, 2023, doi:10.3390/cancers15102816_

Round 1

Reviewer 1 Report

The authors reported an interesting analysis of the role of Netrin family genes in ccRCC. The article is well written and they demonstrated a solid approach to the topic. In the results, they found NTNG1/G2 and NTN4 as possible prognostic biomarkers in patients with ccRCC.

Author Response

Dear Reviewer, 
We would like to thank you for your letter (Manuscript ID: cancers-2301465). 
We appreciate the time you took to review our documents and we are pleased to hear your comments about our work.

Thank you very much again. 
With best regards.

Reviewer 2 Report

The paper by Ke et.al. report findings about Netrin family genes as prognostic markers and therapeutic targets for clear cell renal cell carcinoma where Netrin-4 acts through the 3 Wnt/β-catenin signaling pathway.

This manuscript can has a good potential but the authors first should address one major point before the next round of review. The major flaw of this manuscript is the lack of presenting the rationale behind selection of Netrin family gene as a prognostic marker and/or therapeutic target for ccRCC. As described in the manuscript, the criteria of selection of these genes are not unbiased and with the same rationale multiple similar gene families can be selected as biomarkers or prognostic factors. The authors should first present a robust and verifiable approach for selection of Netrin family as marker and potential targets. They should discuss the roadmap that guided the authors to these genes in an unbiased way. They should also discuss other genes that can be captured in their analysis and potential pitfall of their approach.

Author Response

Response to Reviewers' Comments
Dear Reviewers 
Thank you for your letters and constructive comments on our article (manuscript number: cancers-2301465). All comments have been communicated to all authors, and the manuscript has been carefully responded to based on the reviewers' comments.For the specific reply content, please refer to the attachment
We hope that our response meets your requirements.
Thank you again.
With our most sincere greetings.

Reviewer 3 Report

The authors show the role of netrin4 in renal cell carcinoma, where not many targetable biomarkers are present. They also show that netrin4 is involved in the wnt/b-catenin pathway, potentially opening up therapeutic options. Some of the comments are as follows:

Generally, the authors could provide more details on the methodology and describe the result. For example, do the authors think overexpression causes more apoptosis or cell cycle arrest for the cells? The authors could examine phosphor b-catenin. Did they prepare nuclear extracts for b-catenin localization in the nucleus?

Could the authors provide a link/reference to the HPA-RNA project? It should be mentioned in the data collection section. It can be avoided in the legends later. The GSEs should have references if published. Similarly, R packages should be cited as well.

How does the netrin4 expression compare between HPA and GTEX?

Figures 2A and B seem to be redundant. Is it possible to use only one? How were these figures generated and how many tumor or normal data points were used.

The authors may want to clarify figures 3A/B and E in the legend. I figure 2E2 NTN4 shows a higher HR? does this mean concerning be stage or grade NTN4 high expression is worse prognostically?

Figures 4 and 5 are very confusing and look very similar. The legends have very little information. How is the migration rate defined in figure 6? The animal model metastasis assay could use a better-quality picture since the signal seems pixelated. Could the authors show lung pictures with nodules without the luc overlaps? The authors could point out where the crop was done in the original images.

Author Response

Dear Reviewers 
Thank you for your letters and constructive comments on our article (manuscript number: cancers-2301465). All comments have been communicated to all authors, and the manuscript has been carefully responded to based on the reviewers' comments.For the specific reply content, please refer to the attachment
We hope that our response meets your requirements.
Thank you again.
With our most sincere greetings.

Round 2

Reviewer 3 Report

The authors have satisfactorily answered the queries.